# Photoluminescence Revealed Higher Order Plasmonic Resonance Modes and Their Unexpected Frequency Blue Shifts in Silver-Coated Silica Nanoparticle Antennas

**Atta Ur Rahman** [1] (ID)**, Junping Geng** [1,*] (ID)**, Richard W. Ziolkowski** [2,3] (ID)**, Tao Hang** [1]**, Qaisar Hayat** [1]**, Xianling Liang** [1]**, Sami Ur Rehman** [1] **and Ronghong Jin** [1]

1   Electronic Engineering Department, Shanghai Jiao Tong University, Shanghai 200240, China
2   Global Big Data Technologies Centre, University of Technology Sydney, Ultimo NSW 2007, Australia
3   Department of Electrical and Computer Engineering, The University of Arizona, Tucson, AZ 85721, USA
*   Correspondence: gengjunp@sjtu.edu.cn

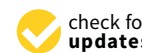

**Featured Application:  Core-shell nanoparticle antennas facilitate the realization of optical metamaterials (engineered artificial materials). Their presence and tunable scattering properties enhance a variety of biological sensing and imaging modalities.**

**Abstract:** Higher order plasmonic resonance modes and their frequency blue shifts in silver-coated silica nanoparticle antennas are studied. Synthesizing them with a wet chemistry method, silica ($SiO_2$) nanoparticles were enclosed within silver shells with different thicknesses. A size-dependent Drude model was used to model the plasmonic shells and their optical losses. Two higher order plasmonic resonances were identified for each case in these simulations. The photoluminescence spectroscopy (PL) experimental results, in good agreement with their simulated values, confirmed the presence of those two higher order resonant modes and their resonance frequencies. When compared with pure metallic Ag nanoparticles, size-induced blue shifts were observed in these resonance frequencies.

**Keywords:** core–shell nanoparticle; nanoantenna; blue shift; higher order modes; size-dependent optical losses

## 1. Introduction

The word nanoantenna is widely used to denote a scattering device able to localize an optical field into a region smaller than its wavelength [1]. An example is a plasmonic-based nanoparticle. The collective oscillation of its surface electrons, i.e., the local surface plasmon resonance (LSPR), produces a local, subwavelength field enhancement in its vicinity. Noble metals like silver (Ag) and gold (Au) exhibit LSPR resonances in optical regime. They are used in a variety of optoelectronics devices for bio sensing, imaging properties, and surface-enhanced Raman spectroscopy wireless communication [2–6].

A variety of different configurations of gold and silver nano-structures, e.g., nanorods, nano-bowties, nanowires, nanoparticles, and nano-cubes have been investigated alone and in combination with dielectrics across the UV-Visible band [5,7,8]. Tuning of the LSPR frequency and/or wavelength of these nanoantennas has been obtained by variations in their size, shape, and material properties [9]. The dielectric constant of the medium in which the metallic nanoparticles are embedded also has a significant impact on their resonance wavelengths [10]. Related experimentally observed LSPR frequency tuning and shifts have been reported. For example, a red shift in the LSPR resonance

wavelength of Ag nanoparticles was reported by Liu et al. [11], when the thickness of their undercoated $SiO_2$ layers was increased. In the same study, a blue shift was also reported when the thickness of an overcoated $SiO_2$ layer was increased. The blue shift was attributed to the increase in the effective dielectric constant of the $SiO_2$ medium when the layer's thickness was increased. Bulk permittivity values were recovered only when that layer became thick. However, when the coating layer was very thin, it was found that the combination of the permittivity values of the air and dielectric yielded the effective dielectric constant.

This article presents the concept of material quantity based (MQB) and nano size-dependent (NSD) optical losses to explain the frequency blue shift observed in fabricated core–shell (Ag-coated $SiO_2$) nanoparticles. Increase in the size of the plasmonic nanostructures under investigation means increasing the amount of lossy materials in the system. Consequently, the LSPR frequency should be red-shifted [12,13]. However, NSD losses mostly occur because of the presence of a larger number of defects that occur in the outer surface of the structure during its nanofabrication. The NSD optical losses are more likely to vanish beyond a certain size limit ($\geq$100 nm) because of smaller surface to volume ratios. In the case of larger particles, the volume effects dominate all other surface issues. Our experiments have identified a blue shift in the LSPRs of the Ag-coated $SiO_2$ nanoparticles. The measured data indicates that an increase in the effective dielectric constant of the $SiO_2$ region is not the only feature responsible for the blue shift. Furthermore, additional size-dependent effects in the nano-regime are necessary to explain the observed blue shifts of the LSPR frequencies.

The Ag-coated $SiO_2$ nanoparticles were synthesized with different Ag shell thicknesses by a wet chemical method. Photoluminescence spectroscopy analyses revealed a blue shift in the LSPR wavelengths. The fabricated nanoparticles were then simulated with the full-wave vector finite element method (FEM) frequency domain solver in CST (Computer Simulation Technology) Microwave Studio. The NSD optical losses were incorporated in the numerical simulations through a lossy Drude model of the Ag layer with a size-dependent collision frequency of the charge carriers. The simulation and experimental analyses demonstrate the existence of higher order resonance modes in all samples. The measured results and their simulated values are in reasonable agreement.

## 2. Materials and Methods

The Ag-coated $SiO_2$ nanoparticles were fabricated using a modified StÖber method [14,15]. A solution of 26.8 mL of ammonium hydroxide (NH4OH) in 200 mL of ethanol was stirred for 30 min at 30 °C in a three-necked round bottomed flask. To grow the monodispersed $SiO_2$ nanoparticles, 4 mL of tetraethyl orthosilicate (TEOS) was quickly added to the mixture and stirred for 10 h. In order to get amino functionalized $SiO_2$ nanoparticles, 0.5 mL of amino-propyl-tri-methoxy silane (APTMS) solution was added to the mixture and stirred for another 8 h. Subsequently, the solution was centrifuged for one hour at 5000 rpm and washed twice with ethanol. After washing the nanoparticles were redispersed in 200 mL of ethanol. A basic solution of silver salt containing potassium was prepared separately by adding 0.05 g of potassium carbonate ($K_2CO_3$) to 200 mL of water, stirred for 15 min to dissolve the $K_2CO_3$ completely, and 2 mL of 1 wt% $AgNO_3$ silver-based solution was added.

In the final step, varying amounts of the amine-functionalized $SiO_2$ solution (1, 2, and 3 mL) were mixed with 8 mL of the K-silver and $K_2CO_3$ solutions to obtain three samples (S1, S2 and S3), each having different Ag shell thicknesses. Solutions were stirred for 10 min. Subsequently, the color of the solutions changed to dark blue when 0.02 mL of formaldehyde was added. The solution was then centrifuged and washed in milli-Q-water for several times to remove unwanted and unreacted chemicals.

The prepared nanoparticles were then dispersed in ethanol with a suitable concentration to prepare the samples for photo-luminance and transmission electron microscopy. A transmission electron microscope (TEM, Model-JEM2100, 200 kV, Japan Electronics Corporation, Tokyo, Japan) was used to obtain information about the sizes, morphology, and thicknesses of the Ag shells. The TEM samples were prepared by the drop coating method, on a copper grid with a carbon film, and then

dried at 150 °C overnight. Finally, the samples were classified on the basis of the measured thicknesses of their Ag shells.

The shell thickness of the observed rough surfaces of the nanoparticles was estimated as follows. First, the thickness of the Ag shells was controlled by the stoichiometry that was used during the experiments, i.e., by the weight of the $AgNO_3$ added to the solution for the shell growth. Second, many nanoparticles were selected throughout the solution and the shell thickness was measured at different locations around each of the chosen ones to obtain an average value. Then, by correlating the outcomes of these two approaches, the reported values of shell thickness were determined.

## 3. Results and Discussion

### 3.1. Transmission Electron Microscopy (TEM) Results

The samples were named as S1, S2, and S3 according to their shell thickness ~14 ± 1 nm, 18 ± 1 nm and 22 ± 1 nm, respectively. The $SiO_2$ core size of all of the samples was ~120 ± 2 nm. TEM images of all of the core–shell samples (S1, S2, and S3) are shown at different magnification in Figure 1, along with those of the bare $SiO_2$ nanoparticles (sample S). The images display uniform size distribution sets of spherical nanoparticles. It was observed that the shells with smaller thicknesses exhibited better surface smoothness as compared to the larger ones.

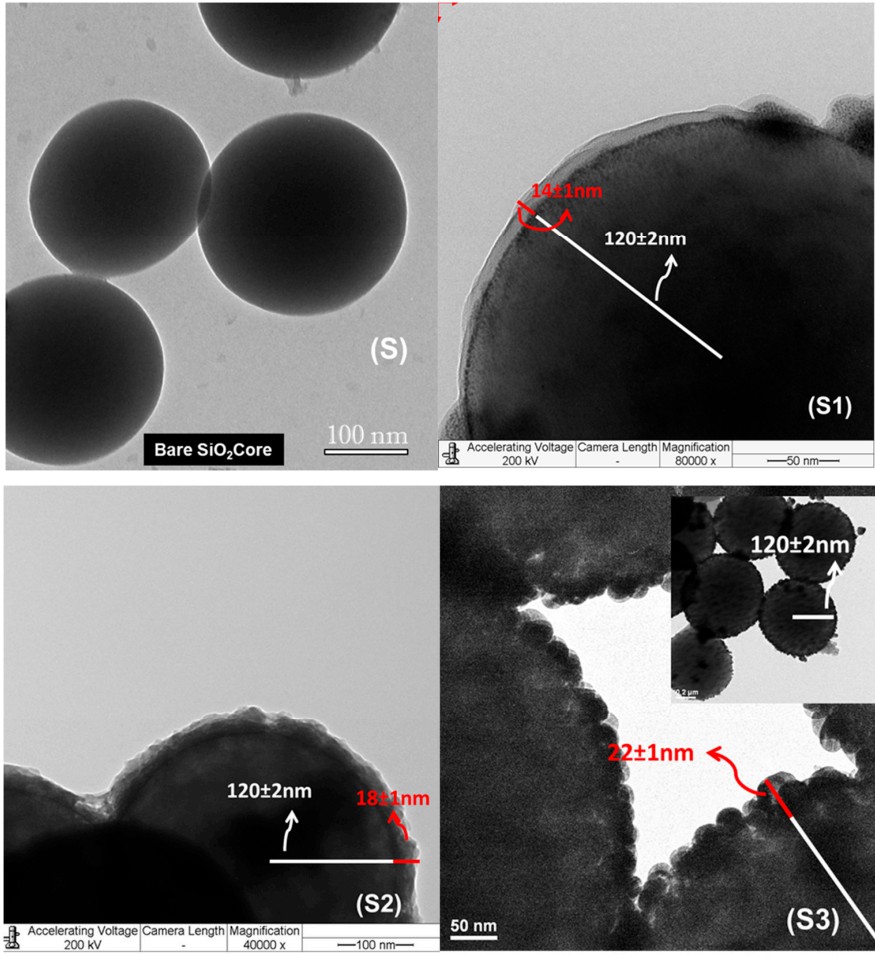

**Figure 1.** TEM images of the sample sets S, S1, S2, and S3. Sample S represents the bare $SiO_2$ nanoparticles. Higher magnification TEM images are given for sets S1, S2 and S3, which have shell thickness of approximately 14, 18, and 22 nm, respectively. The inset in the upper right corner of S3 shows a lower magnification TEM image of the same sample.

### 3.2. Photoluminescence Spectroscopy Analyses

The photoluminescence spectroscopy (PL) analysis was performed by dispersing the nanoparticles in ethanol. The sample was excited with a xenon arc lamp laser emitting in the range from 250 to 850 nm. The response spectra of each sample were recorded from 300 to 650 nm. Figure 2 presents the PL analysis of the three samples S1, S2, and S3, in the form of their absorption intensity as functions of the wavelength of the incident light (electromagnetic, EM) wave. This figure demonstrates that each sample exhibited an optical response consisting of two higher order resonance modes at λ1 and λ2. The corresponding resonance wavelength of each sample falls in the wavelength ranges of λ1: 471–484 nm and λ2: 556–568 nm.

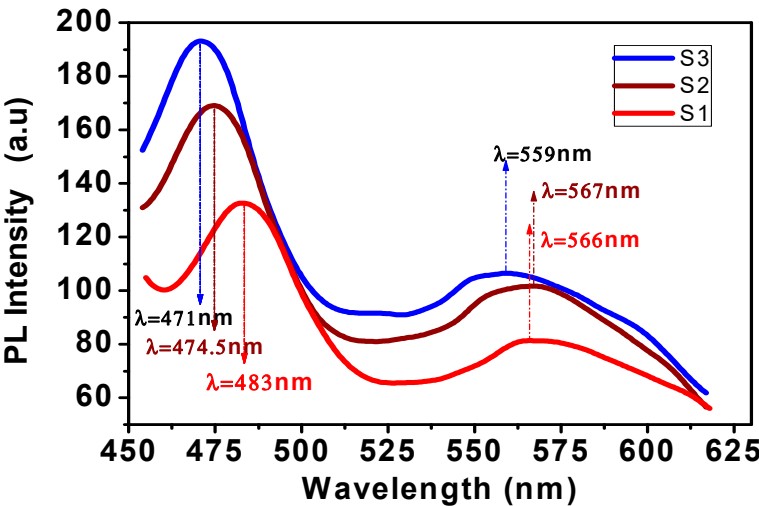

**Figure 2.** The plasmonic response of the three samples S1, S2, and S3, measured by photoluminescence (PL) spectroscopy and their corresponding resonance wavelengths.

The first PL intensity peak for S1 in Figure 2 is $1.3 \times 10^2$ at 483 nm, with the Ag shell thickness being 14 nm. The first PL intensity peak for S2 is $1.7 \times 10^2$ at 474.5 nm, with the Ag shell thickness being 18 nm. The first PL intensity peak for S3 is $1.9 \times 10^2$ at 471 nm, with the Ag shell thickness being 22 nm. It is clear that the first PL intensity peak increases and its position in wavelength moves to smaller values (high frequencies) as the Ag shell thickness increases from 14 to 22 nm. The increase in the intensity of the first peak is due to the decrease in concentration of charge trappers e.g., defects, vacancies, surface oxygen atoms, and dangling bonds, when the shell thickness is increased. This is true for all of the samples. This decrease subsequently leads to a high concertation of oscillating electrons; they are responsible for the observed first large intensity peak of the PL spectra. The corresponding simulations (see Section 3.3 below) incorporate these effects through a size-dependent collision frequency in a Drude model of the Ag shell. A collective blue or red shift occurs with any change in the shell thickness. In particular, the observed blue shift in the frequency response corresponds to an increase in the Ag shell thickness. Similarly, the second peak for S1 is $0.8 \times 10^2$ at 566 nm, the second peak for S2 is $1.0 \times 10^2$ at 567 nm, and the second peak for S3 is $1.05 \times 10^2$ at 559 nm. Again, this behavior shows a similar variance of the PL intensity peak and its wavelength position, i.e., the resonance frequency is blue shifted as the shell thickness increases. These results further confirm that one can indeed reasonably tune the resonance wavelength of a core–shell system by varying the thickness of the Ag shell [11–13].

From the experimental point of view, some factors are very important to consider when the particle size is reduced to the nano-scale. These factors significantly change the real and imaginary parts of the permittivity of the materials. For instance, the surface to volume ratio is very high in the nano-regime. However, most of the atoms lie on the surface of the core. Thus, the surface itself is a more defective region in comparison to the core of the nanoparticle. The atoms on the core have dangling bonds

associated with them. In addition to these dangling bonds, the nanoparticle's surfaces are rich with various types of defects. These include lattice vacancies, extra oxygen atoms, and mismatches between the local lattices. Consequently, the crystallinity of the surface lattice is very weak. These surface lattice defects act as traps for charge carriers opposing the direction of the collective oscillation of the charges, i.e., the plasmons [16].

Because there is an additional interface defined by the boundary of the Ag and SiO$_2$ regions, these effects are stronger in core–shell nanoparticles than in pure metallic ones. Consequently, a depletion layer of charge carriers is formed in the vicinity of this interface [17,18]. This hinders the collective oscillation of the charge carriers associated with the plasmon resonance and results in significant NSD optical and electric losses. The additional kinetic energy of the carriers is thus dissipated in the form of heat. Consequently, the imaginary part of the refractive index (n") and dielectric constant ($\varepsilon$") dramatically increases. The dissipated heat further restricts the collective oscillations of the charge carriers. As the shell thickness decreases, the surface lattice imperfections and interface effects become stronger and lead to significant NSD optical losses. Therefore, a red shift in the resonance frequency occurs with a decrease in the shell thickness. On the other hand, any increase in the shell thickness causes a reduction of the NSD optical losses, ensuring the blue shift is observed in the experiments when the thickness of the Ag-shell increases.

In contrast to the core–shell systems, lattice imperfections and the associated NSD optical losses are negligible in purely metallic nanoparticles. These metal-only nanoparticles have the same type of Ag atoms in both its core and on its surface. Moreover, there are no additional interface effects. This arrangement suggests that the number of surface charge carriers that contribute to the plasmonic response of pure Ag nanoparticles is less than in the core–shell system. It also means that in the plasmonic response of pure Ag nanoparticles, the MQB losses are more prominent in comparison to the NSD optical losses. Therefore, an increase in particle size means an addition of more lossy material. As a consequence, there is always a red shift in pure metallic and dielectric nanoparticles when their size is increased.

### 3.3. Numerical Analysis

To simulate the experimental results, we incorporated the NSD optical losses in the SiO$_2$-Ag core–shell nanoparticles by introducing a thickness dependent collision frequency, Γ(R), in the Drude model introduced to describe the metal. The detailed permittivity model of a passive silver nano-sized shell (≤100 nm), which incorporates the size effects and the interband transitions, was described by Gordon and Ziolkowski [13] and by Johnson and Christy [19]. It is expressed as

$$\varepsilon(\boldsymbol{R}, \omega) = \varepsilon_{Drude}(\boldsymbol{R}, \omega) + \chi_{Interband}(\omega) \tag{1}$$

where $\boldsymbol{R}$ is the thickness of the surrounding shell and the Drude permittivity has the form

$$\varepsilon_{Drude}(\boldsymbol{R}, \omega) = 1 - \frac{\omega_p^2}{\Gamma(\boldsymbol{R})^2 + \omega^2} + j\frac{\Gamma(\boldsymbol{R})\,\omega_p^2}{\omega[\,\Gamma(\boldsymbol{R})^2 + \omega^2\,]} \tag{2}$$

where $\omega_p$ and $\Gamma$ are the plasma and collision frequencies, respectively. The collision frequency becomes especially important in the nano-scale regime because the particle size and mean free path of the charge carriers become comparable and thus, a large resistance to their oscillations occurs. Consequently, both the optical NSD and electrical losses increase substantially. It is known [12,18] that the collision frequency and particle size are inversely related as

$$\Gamma(\boldsymbol{R}) = \Gamma_\infty + \frac{AV_F}{R} \tag{3}$$

For silver at optical frequencies, the terms are $A$~ 1, a constant, and the other Drude parameters and Fermi velocity are m*/m = 0.96, N = $5.85 \times 10^{28}$ m$^{-3}$, $V_F$= $1.39 \times 10^6$ m/s, and $\omega_p$= $1.39269 \times 10^{13}$ s$^{-1}$.

All of the NSD losses described in the PL analysis section were included by incorporating this size-dependent collision frequency into the Drude model. Equation (3) indicates that a decrease in the collision frequency will occur with an increase in the size ($R$) of the nanoparticle. Consequently, the imaginary part of the Drude model (2) will become smaller and the observed blue shift with increasing particle size is demonstrated.

The collision frequency $\Gamma$ can be directly correlated with the NSD optical losses by adjusting its value according to the size of nanoparticle. A file containing the dielectric constant values of Ag, including the NSD losses specified by Equation (2) in the visible frequency range, was imported into the CST simulator. The core–shell nanoparticle was excited with a normally incident plane wave. All of the parameters used in the simulations and those used to fabricate the measured core–shell nanoparticles were the same. In particular, the thickness of the Ag shells for the three samples S1, S2, and S3, were 14, 18, and 22 nm, respectively, and the diameter of the silica core was 120 nm in all cases. The simulation results under these conditions are shown in Figure 3.

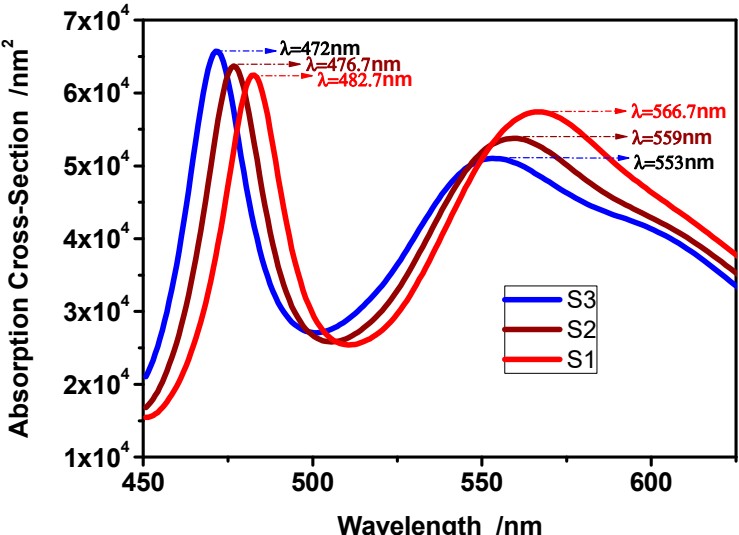

**Figure 3.** The simulated absorption intensity of the core–shell samples S1, S2, and S3, and their corresponding resonance wavelengths.

Figure 3 gives the simulated absorption cross-section (ACS) for the three core–shell cases as functions of the wavelength of the incident plane wave. The results recover the higher order plasmonic resonances observed in the measured results at $\lambda_1$: 470–483 nm and $\lambda_2$: 553–565 nm. Figure 3 indicates that the first ACS peak increases in amplitude as the Ag shell thickness increases from 14 to 22 nm and its position moves to lower wavelengths (i.e., to higher frequencies), which is in agreement with the experimental results in Figure 2. Similarly, the second ACS peak position also moves to higher frequencies as the Ag shell thickness increases, again in agreement with the experimental results in Figure 2. On the other hand, the ACS peak values decrease as the Ag shell thickness increases, in contrast to their measured behavior shown in Figure 2.

Due to large size of the Ag shells, the higher order resonance peaks are quite broad [20]. It is also clear that all of the resonance frequencies are blue shifted as the Ag shell thickness increases. In particular, a direct comparison of the theoretical, simulated and measured resonance frequencies is given in Table 1. The analytical Mie theory, CST numerical and experimental results are in favorable agreement. There are two well-defined peaks in the ACS values within the wavelength range from 450 to 650 nm. The simulation peaks all move to a higher frequency when the silver thickness increases, which is in agreement with the measured outcomes: 14, 18, and 22 nm for S1, S2, and S3, respectively.

**Table 1.** Measured and simulated wavelengths of the peak absorption cross-section (ACS) values.

| Sample | Experiment | | Simulation | | | | | | | |
| | $\lambda_1$ (nm) | $\lambda_2$ (nm) | $\lambda_1$ (nm) | | | | $\lambda_2$ (nm) | | | |
| | | | CST-SDD | Mie-SDD | CST-JC | Mie-JC | CST-SDD | Mie-SDD | CST-JC | Mie-JC |
|---|---|---|---|---|---|---|---|---|---|---|
| S1 | 484 | 568 | 482.7 | 482 | 497 | 496 | 566.7 | 566 | 591 | 588 |
| S2 | 474.5 | 564 | 476.7 | 477 | 469 | 468 | 559 | 557 | 549 | 550 |
| S3 | 471 | 556 | 472 | 470 | 452 | 451 | 553 | 552 | 547 | 545 |

In addition to CST simulation based on size-dependent Drude modal, we also compare the experimental results with CST and Mie theory simulation results based on Johnson & Christy (JC) and size-dependent Drude (SDD) model data shown in Figure 4. In both CST and Mie simulations we used the JC and SDD dielectric data for Silver. It is should be noticed that unlike to the size-dependent Drude model, the Johnson data does not incorporate the additional size effects at nanoscale [19]. A comparison of the normalized, measured CST (JC and SDD) numerical and Mie analytical (JC and SDD) results for Sample S2 is given in Figure 4. It was found that both CST and Mie are coincident, providing sufficient evidence about the reliability of CST to simulate the optical response of core–shell nanoparticles. The resonance frequency points in the CST and Mie simulation are in reasonably close agreement; however, a notable difference with experimental data was predicted using the JC data in both CST and Mie simulation. Obviously this difference is due to the fact that unlike to the Drude model, Johnson model does not include the size effects. However, the simulated results using the SDD data in both cases are reasonably close to the experiment. This further provides evidences for the existence of the NSD and MQB losses and their corresponding blue shift. Confidence in the remaining CST results arises from this comparison and previous applications of it to model nanoparticle phenomena [21] and validation studies with analytical and other numerical approaches [22]. The values of resonance wavelength corresponding to the experiment, CST (JC and SDD), and Mie theory (JC and SDD) are given in Table 1. One can observe from the Table 1 that CST (SDD) and Mie (SDD) simulated results are closer to the experiment as compared to the CST (JC) and Mie theory (JC).

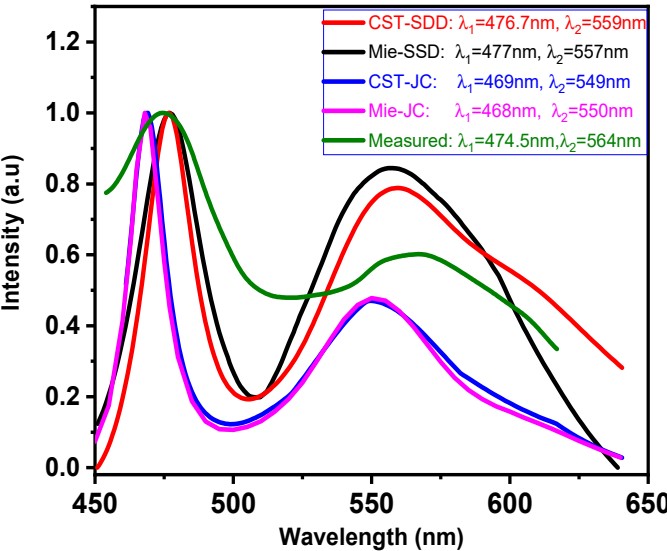

**Figure 4.** Comparison of the measured and simulated (CST and Mie theory) results for Sample S2.

The simulated total electric field distributions in the S2 Ag nanoshell case at its resonance wavelengths are shown in Figure 4. The horizontal panels i and ii in Figure 5 represent the higher order resonance modes at $\lambda_1 = 470$–483 nm and $\lambda_2 = 553$–567 nm, respectively. The vertical panel A gives the corresponding simulated ACS values. The vertical panels B and C represent the corresponding electric field distributions in the YZ and XZ planes at the frequency at which the associated ACS peak

occurs. Note that the incident EM plane wave propagates along the Z-axis from the bottom to the top of the subplots as shown in panel D. The identification of the higher order resonance modes was accomplished by comparing these numerical results with those obtained from Mie theory [21].

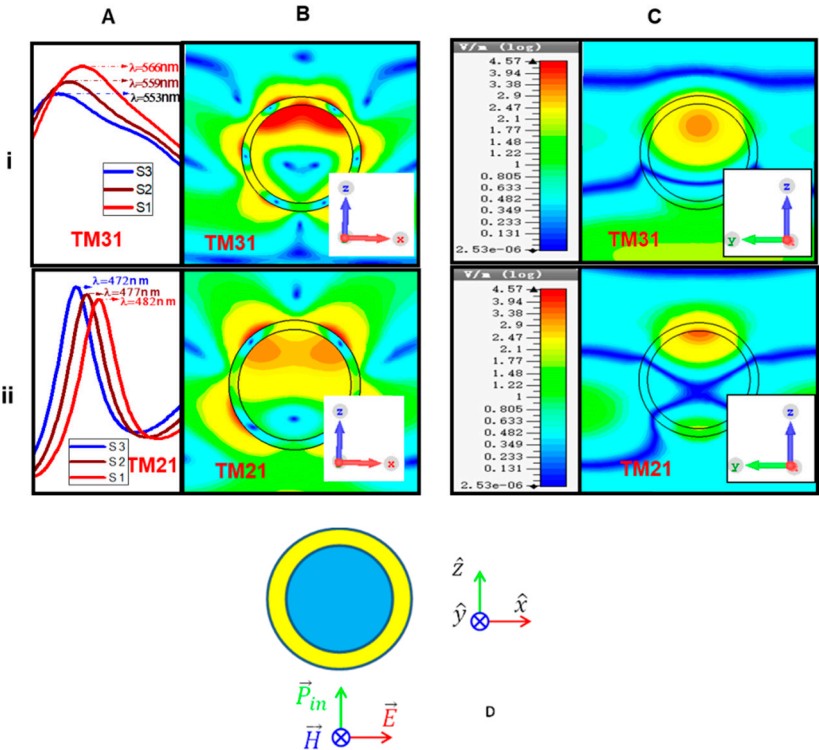

**Figure 5.** (Sample S2) Horizontal panels i and ii represent the higher ordered resonance modes at λ1 = 470–483 nm and λ2 = 553–567 nm, respectively, while vertical panels A, B and C (**A**–**C**) represent the absorption peak, electric field distribution in the YZ and XZ plane of the corresponding resonance modes respectively.

## 4. Conclusions

Sets of SiO$_2$–Ag core–shell nanoparticles with a uniform size distribution were synthesized with a wet chemistry method. The plasmonic response of the Ag-coated SiO$_2$ nanoparticles was simulated, and the results support the experimentally measured PL spectroscopy outcomes. The associated blue shift in the resonance wavelengths of the higher order resonance modes found in the measurements and confirmed with simulation was attributed to the decrease of the NSD optical losses when the thickness of the Ag shell is increased. The MQB optical losses are always dominant in pure metallic nanoparticles and are responsible for the red shift when the size of the pure Ag nanoparticles increases. Both the experimental and simulated results reveal two higher order plasmonic resonances with different wavelength ranges. The numerical and experimental results were compared with the corresponding Mie theory solution. The observed resonance modes were identified as the higher order TM31 and TM21 modes.

**Author Contributions:** A.U.R., J.G., R.W.Z., T.H., and Q.H., designed the experiments and analyzed the measured data. A.U.R. and Q.H. explored the TEM technology and acquired the TEM images. A.U.R., J.G., R.W.Z., Q.H., and S.U.R. made the photoluminescence spectroscopy analyses. A.U.R., J.G., R.W.Z., and S.U.R. performed the analytical and numerical efforts. A.U.R., J.G., R.W.Z., T.H., Q.H., S.U.R., R.J., and X.L. participated in the discussions and supervised the writing of the manuscript. All authors read and approved the final manuscript.

**Funding:** The work was supported in part by the National Natural Science Foundation of China under Grants 61571289 and 61571298, and in part by the Australian Research Council, grant number DP160102219.

**Acknowledgments:** The authors acknowledge Bin Chen from the School of Material Science and Engineering of Shanghai Jiao Tong University for his help in testing and analysis.

**Conflicts of Interest:** The authors declare no conflicts of interest.

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
