# Peer review of "Photoluminescence Revealed Higher Order Plasmonic Resonance Modes and Their Unexpected Frequency Blue Shifts in Silver-Coated Silica Nanoparticle Antennas"

_applsci, doi:10.3390/app9153000_

Round 1

Reviewer 1 Report

This paper demonstrated blue-shift of plasmonic resonances in silver-coated silica nanoparticles (NPs) and proved the size-dependent Drude model by comparing the experimental absorption spectrum and the simulated ones. The understanding of a size-dependent Drude model can be useful in the thin metal coated NP systems which are exploited in the biosensings, nonlinear optical phenomena, and SERS. Since the topic in this paper would be very interesting, the paper is worth to be published if the analysis on the Drude model was properly applied. However, in this paper, the shell size measurement and some analysis should be carefully revised to prove the size-dependent Drude model.

1. In Fig.1, authors should re-review the size measurements on the shell thickness and the core size.

(1) In (S1), (S2), (S3) figures, the shell measurements are not directed along the radial direction. Such angled measurements make the shell thickness to be thicker. If the values, "14nm, 18nm, 22nm" in the figure, are correct, then real shell thickness must be smaller than such values. 

(2) In (S2) figure, the cross-sectional shape of the NP is a ellipse (NOT circle). Long axis seems longer than 120 nm. 

(3) In (S3) figure, the interface between the shell and the core is difficult to identify. Please increase the contrast of the figure. 

(4) In (S2), (S3) figures, the roughness of the shell surface makes the measurement of the shell size difficult. Authors should clarify how the shell thickness is determined by considering the roughness.

2. In the experimental absorption spectrum, there is not information on how to measure PL from the core-shell NPs. Pumping conditions, single NP or assembly of NPs, etc.

3. Authors should explain why the peak intensity at lamda1  increases largely in the experimental spectrum with the shell thickness. In contrast, the simulated ACS spectrum show similar peak heights.

4. In eq. (3), the size dependent Drude model, is there no effect of the shell size on the collision frequency. If there is, please include it in the equation.

5. In Fig. 5, all values and texts are difficult to read.

Author Response

Comment No 1: In Fig.1, authors should re-review the size measurements on the shell thickness and the core size.

Reply: We have reviewed the size measurements associated with Fig. 1 and have corrected any inconsistencies in the revised manuscript.

Comment No 2: In (S1), (S2), (S3) figures, the shell measurements are not directed along the radial direction. Such angled measurements make the shell thickness to be thicker. If the values, "14nm, 18nm, 22nm" in the figure, are correct, then real shell thickness must be smaller than such values.

Reply: The lines inserted in the images were originally drawn simply to highlight the core and shell regions. In response to your comment, we have now drawn these lines according to a fixed scale in the radial direction.

Comment No 3: In (S2) figure, the cross-sectional shape of the NP is an ellipse (NOT circle). Long axis seems longer than 120 nm.

Reply: Thank you for your comment. This was an image processing error. It has been corrected the revised manuscript.

Comment No 4: In (S3) figure, the interface between the shell and the core is difficult to identify. Please increase the contrast of the figure.

Reply: As requested, the contrast of the mentioned figure has been improved and this updated figure has been incorporated in the revised manuscript.

Comment No 5: In (S2), (S3) figures, the roughness of the shell surface make the measurement of the shell size difficult. Authors should clarify how the shell thickness is determined by considering the roughness.

Reply: The shell thickness of the observed rough surfaces of the nanoparticles was estimated as follows. First, the thickness of the Ag shells was controlled by the stoichiometry that was used during the experiments, i.e., by the weight of the AgNO3 added to the solution for the shell growth. Second, many nanoparticles were selected throughout the solution and the shell thickness was measured at different locations around each of the chosen ones to obtain an average value. Then, by correlating the outcomes of these two approaches, the reported values of the shell’s thickness were determined.

We have added these clarifications in the revised manuscript.

Comment No 6: In the experimental absorption spectrum, there is not information on how to measure PL from the core-shell NPs. Pumping conditions, single NP or assembly of NPs, etc.

Reply: Thank you for your comment. The details were in fact addressed already in the original manuscript as:

“The PL analysis was carried out by dispersing the particle in ethanol. The sample was excited by Xenon arc lamp laser working in the range from 250nm to 850nm. The excitation spectra of the sample were recorded from 300nm to 650nm by dispersing the nanoparticles in ethanol.”

Comment No 7: Authors should explain why the peak intensity at lamda1 increases largely in the experimental spectrum with the shell thickness. In contrast, the simulated ACS spectrum shows similar peak heights.

Reply: This is a most relevant and useful comment in regards to main theme of our manuscript. In response, we have included the following in the revised manuscript:

“The increase in the intensity of the first peak is due to the decrease in concentration of charge trappers e.g. defects, vacancies, surface oxygen atoms and dangling bonds, when the shell thickness is increased. This is true for all of the samples. This decrease subsequently leads to a high concertation of oscillating electrons; they are responsible for the observed first large intensity peak of the PL spectra. The corresponding simulations (see subsection 3.3 below) incorporate these effects through a size dependent collision frequency in a Drude model of the Ag shell. A collective blue or red shift occurs with any change in the shell thickness. In particular, the observed blue shift in the frequency response corresponds to an increase in the Ag shell thickness.”

Comment No 8: In eq. (3), the size dependent Drude model, is there no effect of the shell size on the collision frequency. If there is, please include it in the equation.

Reply: In fact, eq. (3) explicitly defines the collision frequency as a function of the shell thickness, i.e., as defined immediately after eq. (1), the quantity R.

Comment No 9: In Fig. 5, all values and texts are difficult to read.

Reply: The contents of the mentioned figure have been improved and made more readable in revised manuscript.

The image has been modified and its updated version included in the revised manuscript.

Reviewer 2 Report

In the paper “Photoluminescence Revealed Higher Order Plasmonic Resonance Modes and Their Unexpected Frequency Blue Shifts in Silver-Coated Silica Nanoparticle Antennas” by Atta Ur Rahman et al., the Authors study plasmonic losses in silver-silica core-shells. The core-shell particles with different shell thickness are fabricated by a wet chemical approach. The optical response of the fabricated particles is studied via PL measurements. The results are compared with numerical simulation results of absorption cross section based on the size-depended Drude model. The numerical and experimental results are in good agreement. Overall, I think that the paper is well written and presented results would be interesting to nanophotonics community. I have only one comment:

1. The legend in Figure 4 – It says “Mie-SSD”, should it be “SDD” instead? And if it is the case, why the numerical results and Mie results are slightly different. Mie theory is rigorous and CST results must be exactly the same. 

Author Response

The legend in Figure 4 – It says “Mie-SSD”, should it be “SDD” instead? And if it is the case, why the numerical results and Mie results are slightly different. Mie theory is rigorous and CST results must be exactly the same.

Reply:  Thank you. The word “Mie-SSD” refers to the fact that the Ag shell size effects have been included in the Mie theory calculations and, consequently, in the associated results. The notation in Figure 4 is correct.

Because the Mie theory is an exact analytical approach while CST is strictly a full wave numerical approach, they will never be precisely identical until the level of discretization approaches the continuum limit.

Round 2

Reviewer 1 Report

The revised paper included proper answers for the issues raised by the reviewer. I recommend this paper is now worth to be published as it is.